# General Practitioners’, Pharmacists’ and Parents’ Views on Antibiotic Use and Resistance in Malta: An Exploratory Qualitative Study

**DOI:** 10.3390/antibiotics11050661

**Published:** 2022-05-14

**Authors:** Hager Ali Saleh, Michael A. Borg, Cecilia Stålsby Lundborg, Erika A. Saliba-Gustafsson

**Affiliations:** 1Department of Global Public Health, Health Systems and Policy (HSP): Improving Use of Medicines, Karolinska Institutet, Tomtebodavägen 18A, 171 77 Stockholm, Sweden; hager.saleh@stud.ki.se (H.A.S.); cecilia.stalsby.lundborg@ki.se (C.S.L.); 2Department of Infection Prevention and Control, Mater Dei Hospital, MSD 2090 Msida, Malta; michael.a.borg@gov.mt; 3Faculty of Medicine and Surgery, University of Malta, MSD 2080 Msida, Malta

**Keywords:** antibiotics, antibiotic resistance, parents, public, general practitioners, pharmacists, respiratory tract infections, qualitative, focus group discussion

## Abstract

Background: Antibiotic resistance (ABR) remains a global health threat that requires urgent action. Antibiotic use is a key driver of ABR and is particularly problematic in the outpatient setting. General practitioners (GPs), the public, and pharmacists therefore play an important role in safeguarding antibiotics. In this study, we aimed to gain a better understanding of the antibiotic prescribing-use-dispensation dynamic in Malta from the perspective of GPs, pharmacists, and parents; Methods: we conducted 8 focus groups with 8 GPs, 24 pharmacists, and 18 parents between 2014 and 2016. Data were analysed using inductive and deductive content analysis; Results: Awareness on antibiotic overuse and ABR was generally high among interviewees although antibiotic use was thought to be improving. Despite this, some believed that antibiotic demand, non-compliance, and over-the-counter dispensing are still a problem. Nevertheless, interviewees believed that the public is more accepting of alternative strategies, such as delayed antibiotic prescription. Both GPs and pharmacists were enthusiastic about their roles as patient educators in raising knowledge and awareness in this context; Conclusions: While antibiotic use and misuse, and knowledge and awareness, were perceived to have improved in Malta, our study suggests that even though stakeholders indicated willingness to drive change, there is still much room for improvement.

## 1. Introduction

Antibiotic resistance (ABR) remains a threat to global health for which urgent action is needed, lest we end up in a post-antibiotic era [1]. In Europe alone, antibiotic-resistant infections cause the deaths of approximately 33,000 people annually [2]. ABR is however a multisectoral and multifaceted problem, influenced by multiple factors and different stakeholders [3,4]. Strategies to tackle ABR must therefore address issues across various societal and health system levels, and change behaviour among numerous stakeholders, including healthcare professionals, the public, and agricultural workers [1,5,6].

One key driver of ABR is antibiotic use, as is evident in southern European countries where antibiotic consumption and resistance rates are among the highest in Europe [7,8,9,10]. In this setting, most antibiotics prescribed for systemic use are provided in the outpatient setting [11], with over half of prescriptions often unwarranted or inappropriate [12]. This behaviour is influenced by several factors, including clinical, patient, environmental, and cultural factors, such as patient demand and expectations, knowledge and awareness, and provider attitudes and characteristics [13,14,15,16,17]. Antibiotic consumption is also driven by unprescribed antibiotic use through for example, self-medication, medicine sharing, and over-the-counter dispensing [18,19]. In fact, in 2011, an estimated 19–100% of antibiotics used outside northern Europe and North America were consumed without a prescription [19], which led several countries to prohibit non-prescription use [19,20]. Nevertheless, over-the-counter dispensing persists in some countries [21].

Repeated Eurobarometer surveys over the past decade have suggested that Malta, a southern European country, has some of the highest antibiotic consumption rates in Europe [7,22,23]. In 2018, 42% of Maltese reported having consumed antibiotics the previous year (surpassed only by Italy at 47%), predominantly for sore throat and flu, most of which were the result of a medical prescription [7]. Over-the-counter use, however, has declined [24]. Local general practitioners (GPs) however, claim that over-the-counter dispensation by pharmacists is a problem in Malta [5], although data show that non-prescription use has reduced substantially [24]; in 2018, it was the fourth-lowest in the EU/EEA [7]. GPs also reported that self-medication and patient demand is a problem in Malta [5]. Surveillance data however, suggests that patient demand is low [25], indicating that this may also be overestimated by GPs [26,27]. General public knowledge on appropriate antibiotic use in Malta was also below average among European countries [7].

Since most antibiotic consumption in Malta occurs in the community [28], GPs, the public, and pharmacists all play an important role. Therefore, we aimed to gain a better understanding of the antibiotic prescribing-use-dispensation dynamic in Malta from the perspective of three stakeholders: GPs, community pharmacists, and parents, with a focus on respiratory tract infections (RTIs). Specifically, we aimed to: (i) explore stakeholders’ understanding of prudent antibiotic use and ABR and their reported practices, (ii) develop a deeper understanding of how the ‘client-GP-pharmacist’ dynamic influences antibiotic use, and (iii) identify drivers of antibiotic use and potential solutions to address ABR.

## 2. Materials and Methods

### 2.1. Setting

In Maltese primary care, patients are not registered with a particular GP. About two-thirds of primary care is delivered by private sector GPs [29] and patients pay out-of-pocket; no subsidy or reimbursement applies. Private GPs frequently practice within retail pharmacies or private clinics, and home visits are still popular [30]. In the public sector, GP clinics are walk-in clinics and services are free of charge to residents. Antibiotics are prescription-only medicines by law, primarily obtained out-of-pocket from community pharmacies [24]. Prescriptions are valid for 10 days after issue [31]. Additionally, pharmacists should retain single-use prescriptions after dispensing [32].

### 2.2. Data Collection and Analysis

This study made part of the pre-intervention phase of a larger project, the Maltese Antibiotic Stewardship Programme in the Community (MASPIC) [33]. We conducted 8 focus group discussions (FGDs) with three stakeholder groups; GPs, community pharmacists, and parents of children under 12 years. Semi-structured FGD guides were developed by the research team and informed by a thorough literature search and co-authors’ subject-area knowledge and local expertise (Appendix A). Areas for discussion included: antibiotic use and resistance, health-seeking and antibiotic dispensing behaviour, stakeholder interaction, role is combating ABR, and suggestions for improvement.

A purposive sample of GPs registered on the Malta Medical Council’s Register and who had previously participated in individual interviews about antibiotic use and resistance, were called and invited to participate in a FGD in November 2014. Eleven GPs accepted to participate and were split into two focus groups. Three GPs, however, dropped out last minute from the 2nd focus group, therefore the final sample consisted of 8 GPs (*n* = 6 and *n* = 2, respectively). FGDs with GPs were held in the evening at a central hotel, and GPs were offered a light dinner as compensation for their time.

To incorporate perspectives of multiple stakeholders, FGDs were later also held with pharmacists and parents of children under 12 years in May 2016. Eligible pharmacists were identified through the local pharmacy register and recruited using snowball sampling. Pharmacies across the island were called and practising pharmacists were invited to participate. They were also asked to recommend up to three fellow pharmacists to participate in the study. Twenty-four pharmacists agreed to participate and divided into three different focus groups (*n* = 7, *n* = 11, and *n* = 6, respectively). Finally, a snowball sample of eligible parents were invited through schools and asked to recommend up to three other parents to participate. Ultimately, three FGDs were held with 18 parents (*n* = 5, *n* = 5, and *n* = 8, respectively) (Table 1). All FGDs with pharmacists and parents were held in the evenings at the local governmental hospital and participants were offered a light dinner as a token of appreciation.

E.A.S.-G. held FGDs with an observer/note-taker that lasted 1.5–2 h/group. Sessions were audio recorded following consent from all participants and transcribed verbatim. Data were analysed iteratively by E.A.S.-G. and H.A.S. using deductive and inductive content analysis [34,35]. Data were also triangulated (data source triangulation) to better understand stakeholders’ perspectives. In brief, transcripts were read individually several times to acquire a sense of the whole. Meaning units were selected and shortened into condensed meaning units, from which codes were derived. Similar codes were grouped into sub-categories and coalesced into categories. Analysis was frequently discussed by E.A.S.-G. and H.A.S., and later with the remaining co-authors, until consensus was reached.

### 2.3. Ethical Considerations

Ethical approval was sought from the University of Malta’s Research Ethics Committee (UREC) on two occasions. For the GPs’ FGDs, the study was deemed exempt. UREC granted ethical approval for the pharmacists’ and parents’ FGDs. Standard ethical protocols were adhered to during all FGDs. All participants were informed about the study’s purpose and that they were free to withdraw from the study or decline to answer questions without consequence. Informants were also given the opportunity to pose queries about the study, following which signed informed consent was obtained.

## 3. Results

Our findings are organized and presented across three overarching categories: (1) antibiotic use and ABR in Malta from the perspective of three stakeholder groups, (2) influence of interpersonal relationships among patients, GPs, and pharmacists on antibiotic use, and (3) solutions for action—tackling ABR in Malta.

### 3.1. Category 1: Antibiotic Use and ABR in Malta from the Perspective of Three Stakeholder Groups

#### 3.1.1. Perceptions on ABR and Attitudes towards Antibiotic Use in Malta

GPs, pharmacists, and parents reported that antibiotics are overused and misused in Malta, and that consequently antibiotics are losing their effectiveness. In addition, they noted that ABR has become problematic, not just in the community but also other sectors such as agriculture.

“We are moving towards a time and an age where it will be very difficult to treat a normal respiratory infection because bacteria are getting very resistant.”(FGD5—pharmacists)

GPs and pharmacists opined that although some people harbour misconceptions and believe antibiotics are “the holy grail”, public awareness has improved. GPs remarked that patients are more accepting of alternative strategies such as delayed antibiotic prescribing (DAP) and symptomatic therapy without antibiotics, particularly if coupled with patient education.

“…there is less pressure for us to prescribe and much more acceptance by people to accept either delayed antibiotic therapy or symptomatic therapy…”(FGD1—GPs)

GPs and pharmacists also explained that some people question antibiotic prescriptions or refuse them altogether, particularly parents for their children, as corroborated by interviewed parents. This was not received well by all GPs, who found it hard to persuade such patients when antibiotics are recommended. Nevertheless, both pharmacists and GPs believed that, although improving, patient demand remains a problem, and both admitted to occasionally succumbing to the pressure. According to pharmacists and GPs, some patients expect antibiotics and are disappointed when they do not receive them. Others threaten to acquire antibiotics elsewhere, which they believed may impact their reputation.

“…there are those who tell you, “I came to get antibiotics”, and when you tell them they don’t need they keep insisting because the child has fever.”(FGD2—GPs)

GPs and pharmacists believed that insufficient knowledge on appropriate management of cold and flu partly explains why patients pressure doctors. Patients’ expectation of instant relief was also believed to contribute to the problem, particularly if they must return to work or send their children to school. This was corroborated by parents who admitted that they sometimes do not afford taking sick leave and therefore expect antibiotics for their children.

“Especially those who are either self-employed or have children who want to be sent to school early… they cannot afford to be on sick leave, so they ask for antibiotics.”(FGD1—GPs)

In contrast, some parents remarked that GPs sometimes prescribe antibiotics despite only expecting a sick leave certificate.

“… I go because I feel very sick and am unable to go to work and need a certificate. … A lot of times he gives me antibiotics and I challenge myself not to take them…”(FGD6—parents)

#### 3.1.2. GPs’ Antibiotic Prescribing Practices

Pharmacists reported that GPs’ antibiotic prescribing practices vary but that older GPs exhibit habitual prescribing behaviour, relying on experience more than guidelines, whereas younger GPs are more guideline-concordant. Consequently, some pharmacists stocked their pharmacy based on local doctors’ preferences.

“…the older generation of doctors prefer certain types of antibiotics… I have a new GP in the pharmacy, who sticks to guidelines quite a lot…she does not have a habitual prescribing preference.”(FGD3—pharmacists)

While parents agreed that GPs prescribe habitually, they believed that GPs overprescribe irrespective of age. Nevertheless, pharmacists and GPs believed that unnecessary antibiotic prescribing has declined, although they agreed that antibiotics are still regularly prescribed for viral infections or as an attempt to prevent secondary bacterial infections. They attributed this partly to a perceived lack of diagnostic support and local community resistance data. They believed that diagnostic uncertainty and fear stemming from prior negative experiences, further drives unnecessary and broad-spectrum antibiotic prescribing. Finally, GPs mentioned occasionally prescribing combination antibiotic therapy to cover both Gram-positive and Gram-negative bacteria, and ensure swift patient recovery, which was corroborated by pharmacists.

“…when the patient needs to go to work or school quickly, I would give two antibiotics, not one… to cover Gram-negative and positive bacteria.”(FGD1—GPs)

#### 3.1.3. Delayed Antibiotic Prescribing (DAP)

All interviewed stakeholders were generally in favour of DAP and its ability to reduce unnecessary antibiotic consumption and patient follow-up, and over-the-counter dispensing. GPs mentioned that they provide delayed antibiotic prescriptions in specific use-cases and that patients are becoming increasingly accepting of the strategy. Pharmacists reported that DAP practices varied by locality and that it seemed more popular among younger doctors and for use in children.

“…it [DAP] reduced a lot of antibiotics that are used and has released a lot of pressure on us to give antibiotics over-the-counter.”(FGD3—pharmacists)

Pharmacists added that clients seldom insist on dispensing delayed antibiotics prematurely without a valid reason, e.g., travel or difficulty accessing a pharmacy at night or during the weekend. Despite increasing acceptance of DAP, some pharmacists believed it is suboptimal to shift decision-making onto the patient and were therefore less in favour of it.

“I don’t agree with it because they leave it at the discretion of the patient. …if the patient doesn’t need antibiotics, then don’t prescribe.”(FGD3—pharmacists)

Indeed, whilst most parents stated they would unlikely consume a delayed antibiotic themselves, when it came to their children, they felt uncomfortable with the responsibility-shift. As a result, some parents preferred to have the antibiotic at hand just in case or that the GP see their child again. GPs however, believed that patients may perceive them as avaricious if they advised patients to return.

“…if it is for my children … I prefer if they check them again because let’s say another two days have passed, I can’t check his throat. I don’t know if it’s better or worse… If he gives me the prescription and tells me in two days buy them, I would buy them now.”(FGD7—parents)

Consequently, GPs found combining DAP with advice to call back if symptoms persist to be a good compromise, and parents expressed they would be more accepting of DAP if doctors provided better guidance. Pharmacists also believed that DAP could work if combined with patient education, and both pharmacists and GPs said they used these encounters as an educational opportunity on appropriate antibiotic use.

Other potential drawbacks of DAP noted by pharmacists included the GP’s inability to confirm whether a patient took the antibiotic or not. They also believed that the customary two-day delay is insufficient as symptoms are unlikely to resolve so quickly. Pharmacists and GPs also described that GPs’ DAP methods varied, e.g., some forward-dated or marked prescriptions with a cross. Pharmacists familiar with local GPs’ practices understood these varying methods but noted that other pharmacists may not understand “the system”.

“…the GP marks it with a cross. I understand her but if the patient goes to another pharmacy, I am not sure they [pharmacists] would understand.”(FGD3—pharmacists)

#### 3.1.4. Over-the-Counter Antibiotic Dispensing

Most pharmacists reported that they do not dispense antibiotics over-the-counter, and that patients generally accept this, especially if coupled with a rationale. Parents agreed that antibiotics are rarely dispensed over-the-counter. GPs’ views were mixed however, with some stating that it has not ceased but is on the decline, particularly since the introduction of indemnity insurance. Both pharmacists and parents admitted that in a small setting such as Malta, over-the-counter dispensing happens more frequently when people know one another.

“…we’re in a small island where everyone knows one another so using the prescription as a barrier in Malta is less effective.”(FGD4—pharmacists)

“...it is not the first time that I just went and bought the medicine, I know that she shouldn’t give it to me over-the-counter, but she got used to me and gives it to me.”(FGD6—parents)

#### 3.1.5. Compliance and Self-Medication

Both GPs and pharmacists noted that patients (including themselves) are often non-compliant, self-medicate, store antibiotics at home (likely from previously discontinued antibiotic courses), and share antibiotics with family and friends.

“I have the issue of compliance myself and I know how important it is to finish antibiotics, but I never do, let alone patients.”(FGD4—pharmacists)

Parents admitted to self-medicating, although rarely, but that they would not give their children antibiotics without first consulting a doctor. Similarly, while parents completed their children’s antibiotic courses, they reported sometimes stopping their own course early of their own accord, to limit antibiotic treatment or because they felt better.

“…if I feel better, I stop. I know it’s wrong because I’m supposed to have the full course, but if I can avoid them, I stop them.”(FGD6—parents)

#### 3.1.6. Antibiotic Disposal Practices

Pharmacists reported that people dispose antibiotics inappropriately and are generally unaware about how to dispose of them safely. In fact, parents did not know where to dispose antibiotics safely and admitted to disposing leftover antibiotics inappropriately (e.g., flushing down the toilet). Pharmacists acknowledged their role in supporting safe antibiotic disposal with some reporting that they personally dispose of clients’ leftover antibiotics at local disposal sites. Pharmacists suggested placing disposal bins in pharmacies to encourage appropriate disposal and create awareness.

“They [patients] flush antibiotics down the toilet … it’s the worst thing you can do.”(FGD3—pharmacists)

### 3.2. Category 2: Influence of Interpersonal Relationships among Patients, GPs, and Pharmacists on Antibiotic Use

#### 3.2.1. The GP-Patient and Pharmacist-Client Relationship

GPs mentioned that patients’ quick access to GPs could contribute to antibiotic overconsumption.

“The advantage—and maybe a disadvantage—that we have in Malta is that the patient feels sick and makes contact with the doctor on that same day. […] We tend to see the disease very early on and most of the time, it’s difficult to decide, with the few signs that you have, what is going to happen to the patient. So, there may be a high tendency to give the antibiotic.”(FGD2—GPs)

GPs also believed that patients select their GP based on the GP’s willingness to prescribe. This was corroborated by parents, although when it came to their children, parents seemed to trust GPs with judicious antibiotic prescribing more. Conversely, pharmacists explained that some people believed good doctors prescribe antibiotics whereas less competent doctors prescribe less antibiotics.

“I don’t take them [children] to my GP because he immediately prescribes antibiotics”(FGD3—parent)

Pharmacists viewed themselves as trusted medical professionals, from whom clients seek advice, although they reported that clients’ trust in pharmacists varied. Interestingly, parents remarked that pharmacists are better communicators than physicians.

“…there are those who just take the doctor’s opinion and consider only that and the pharmacist doesn’t exist but now there are a lot who want the pharmacist’s opinion first…”(FGD3—pharmacists)

Pharmacists added that patients’ trust in the doctor, a positive interaction with a pharmacist, and sufficient information about possible side effects and the importance of finishing the antibiotic course, improves compliance.

#### 3.2.2. The GP-Pharmacist Relationship

The nature of the GP-pharmacist relationship varied from highly collaborative to rather detached. Pharmacists typically engaged in more open communication and collaboration with younger GPs, GPs who worked locally or at the same pharmacy.

“…certain doctors, especially young doctors, they are very understanding and can relate, but especially doctors that are used to working alone, it’s like they feel insulted that I call.”(FGD5—pharmacists)

Pharmacists described frequently calling GPs to discuss alternative treatment options or request clarification on prescriptions, for example. Similarly, GPs sometimes requested prescription advice from pharmacists. Some pharmacists suggested that encouraging better collaborative relationships between and among GPs and pharmacists would be beneficial.

### 3.3. Category 3: Solutions for Action—Tackling ABR in Malta

Tackling ABR was perceived to be: “…the responsibility of everyone—the patient, doctor, pharmacist, national systems…” (FGD3—pharmacists). Interviewees mentioned several potential solutions that could be implemented in this context, including educational campaigns, data access, guidelines and diagnostic support, and policy changes.

#### 3.3.1. Education and Nationwide Campaigns

GPs and pharmacists agreed that they play a critical role in educating patients on appropriate antibiotic use, drug interactions, side effects, and compliance.

“...we cannot just say that antibiotics are not good for viruses. Why are they not good? What is the consequence? You cannot tell them, “I’m not going to give you antibiotics because of the resistance” because what is resistance?”(FGD5—pharmacists)

GPs believed that individual patient education is more effective than public campaigns; written instructions were particularly helpful. Parents expected their GPs to provide a rationale for prescribing (or not prescribing) antibiotics but acknowledged that this did not always occur. In fact, pharmacists remarked that GPs spend insufficient time educating patients due to time constraints. They added that education is challenging when the pharmacy is full.

“...you will always have a sector of the population where no matter how much time you spend trying to educate them [through public campaigns], you will not deliver the point. So, to some extent patient education must be from us—an individual approach at our clinic.”(FGD1—GPs)

In addition to patient-level education, pharmacists and parents believed that ongoing, targeted nationwide campaigns delivered by experts in the field are needed. Pharmacists also suggested targeting schools, and mother and baby classes.

“…to get the message through it must be a mass thing not an ‘antibiotic month’ only. You must have doctors and representatives from the health department on key radio and television channels and different populations must be targeted in different ways.”(FGD5—pharmacists)

Finally, GPs were also in favour of GP education. However, they acknowledged that given their long working hours, attendance might be challenging. Pharmacists agreed and suggested that alternate strategies are needed to target GPs who cannot or are reluctant to attend education sessions.

#### 3.3.2. Data access, Guidelines, and Diagnostic Support

GPs believed that making local community data available through accessible national guidelines is important to tackle ABR. Although GPs reported that lack of diagnostic support limited their ability to make informed prescribing decisions, they believed that in the private sector, the added cost of diagnostic testing would not be well-accepted by patients.

Pharmacists recommended introducing electronic medical records in the community nationwide to support GPs’ prescribing decisions. Without medical records, pharmacists believed that GPs are more likely to prescribe habitually rather than tailoring prescribing to the individual.

“…practically all doctors don’t keep notes. So, they cannot observe patterns, like, “This patient always has a tendency of having fever that spikes quickly, so there is no need of antibiotics”. […] That, I think, is a big problem.”(FGD5—pharmacists)

Similarly, pharmacists also suggested introducing e-prescriptions to track prescriptions and monitor GPs’ prescribing, which may inadvertently influence GPs’ prescribing. E-prescriptions would also eliminate illegible prescriptions that often lack sufficient information.

#### 3.3.3. Policy Changes

Finally, pharmacists encouraged changing sick leave policies to allow parents to claim sick leave when their children are ill and discourage sending them to school or childcare when ill. GPs also wished for enforcement of regulations on over-the-counter antibiotic sales, to cease the practice altogether.

## 4. Discussion

### 4.1. Summary of Key Findings

In this study, we triangulated the views on antibiotics and ABR of three important stakeholders: GPs, pharmacists, and parents of young children. Awareness that antibiotics are being overused and that ABR is increasingly becoming problematic was generally high among interviewees although antibiotic use was thought to be improving in Malta. Despite this, some participants believed that antibiotic demand, non-compliance, and over-the-counter dispensing is still a problem. While over-the-counter dispensing has improved [24], Malta has only seen marginal improvement in knowledge and awareness of appropriate antibiotic use. As noted by interviewees, awareness and understanding of antibiotics remains below European average, with many still believing that antibiotics kill viruses and are effective against colds [7]. Nevertheless, interviewees mentioned that the public is more accepting of alternate strategies, including DAP. GPs and pharmacists alike had a very positive outlook on their role as patient educators, to raise knowledge and awareness in this context. Remaining areas for improvement raised were related to appropriate antibiotic disposal practices, and better access to guidelines, diagnostic support, and local community resistance data. It is worth noting that although local community resistance data has been reported every year [36], GPs did not seem aware of it indicating the need to better spread these data to community prescribers.

### 4.2. Demand, Expectations, and Sick Leave Policies

Despite a perceived improvement in awareness, patients were still reported to demand antibiotics. Pharmacists and doctors also perceived that; patients expect antibiotics. Antibiotic demand is a key modifiable driver of unnecessary antibiotic use and remains a problem in various contexts [13,27,37,38]. When faced by such pressure, doctors are more likely to prescribe antibiotics to meet patient expectations. Similarly, doctors’ perception that patients expect antibiotics, can lead to significantly higher unnecessary antibiotic prescribing [26,27,37,39]. Indeed, interviewed parents noted mismatched expectations between them and their doctor, who sometimes prescribed antibiotics despite them expecting otherwise.

In our study, demand was believed to be fuelled by many factors, including the public’s subpar knowledge, the need for instant symptom relief, and stringent sick leave policies. Some parents admitted they expected antibiotics for their children when unable to take sick leave, so they could return to school sooner. At the time of this study, a sick leave certificate issued by a medical doctor was required from the first day of illness if social security benefits were to be claimed after three days of illness. Such a demand, encourages early consultation with GPs, which in turn could increase the odds of premature initiation of antibiotic treatment, as has been seen in other countries where sick leave certificates are required when absent for <7 days [40]. Requesting certificates of absence from school so early could have similar repercussions. Compounding the problem, was the lack of social support for parents with sick children who are unable to return to school due to illness. We advocate for the revision of sick leave policies, including providing parents the opportunity to be on family leave when their children are unwell.

### 4.3. Delayed Antibiotic Prescribing (DAP)

Interviewees noted that DAP is increasingly being accepted in this setting, although not all agreed with this strategy. The predominant reason was hesitancy to transfer decision-making responsibility onto patients. This reluctance has been shown elsewhere [41,42,43,44]. In fact, most parents confirmed that they would rather have their children seen by a doctor again than decide themselves whether antibiotics are necessary. However, with better medical guidance, parents reported that they would more readily accept a delayed prescription. Both pharmacists and GPs agreed that DAP could be used in conjunction with patient education. Delayed prescriptions can in themselves educate and empower people while also establishing trust and connection with the doctor [45].

Pharmacists also raised other issues with DAP in Malta, including lack of standardized delayed prescriptions that resulted in doctors employing varying methods, which could lead to miscommunication among all stakeholders. Pharmacists also perceived that a two-day delay was too short, particularly as many viral RTIs may need 1 to 4 weeks to resolve, depending on the complaint [46]. Clearer guidance on DAP and standardization of practices is therefore highly suggested in this context. Finally, pharmacists found the inability to track patients’ compliance with delayed antibiotic prescriptions concerning. They proposed introducing e-prescribing to eliminate illegible prescriptions, track prescriptions, monitor GPs’ prescribing patterns and possibly influence GPs’ prescribing behaviour. E-prescribing could improve antibiotic prescribing quality [47], and help further minimize over-the-counter antibiotic dispensing, by offering the opportunity to monitor prescribing and dispensing trends [48], which is much-needed in this setting.

### 4.4. Patient Education and Communication

GPs believed that tailored patient education is more beneficial than public efforts. Indeed, studies have shown that antibiotic campaigns intended to improve knowledge and consequently behaviour, have had mixed effects [49,50]. Nevertheless, both pharmacists and parents believed that, in addition to individual patient education, continued national campaigns delivered by experts in the field are required. Conversely, interviewed GPs found providing written instructions to patients particularly useful. Written instructions can significantly decrease antibiotic use without impacting patient satisfaction [51]. In fact, parents indicated a desire for GPs to better explain their prescribing rationale.

Involving community pharmacists in raising awareness through patient education could also have a positive impact [52]. Pharmacists have an integral role in the healthcare system. As frontline healthcare professionals, community pharmacists provide convenient access to a qualified medical professional and therefore play a crucial role in combating ABR. In our study, pharmacists viewed themselves as trusted medical professionals from whom clients sought advice but noted that clients’ trust in them varied. Indeed, according to the 2018 Eurobarometer report, doctors are the most trusted source of information on antibiotics in the Maltese context [7]. Interestingly, interviewed parents stated that pharmacists were better communicators than physicians. Indeed, pharmacists emphasized that they routinely explained when antibiotics are or are not necessary, how they should be taken, and how to safely dispose of them. Providing individualised information about antibiotic duration, dosage, and regimen, has been shown to improve knowledge and adherence to antibiotic treatment [50,53]. Undoubtedly, pharmacist involvement in antibiotic stewardship should not be overlooked, particularly in contexts such as Malta, where over-the-counter antibiotic dispensing is low.

In short, promoting and facilitating tailored patient education through both GPs but also pharmacists could help address knowledge gaps and align expectations, and improve modifiable behaviours such as demand, self-medication, and non-compliance. It could also support greater patient involvement in their own care planning. However, although imparting knowledge and raising awareness through education is an important component of behaviour change, it is not sufficient to change behaviour on its own [54,55], and must be coupled with other behaviour change strategies and antibiotic stewardship activities.

### 4.5. Interdisciplinary Collaboration

Both pharmacists and GPs reported that GPs occasionally asked pharmacists for prescription guidance, and some pharmacists believed that improving collaboration between GPs and pharmacists would be advantageous. Although community pharmacists’ attitudes towards their involvement in antibiotic stewardship activities and their potential impact on GPs antibiotic prescribing have seldom been explored [56,57], studies indicate they can promote guideline-concordant antibiotic prescribing among GPs and reduce antibiotic prescribing [58]. Pharmacists could also play a more active role in screening and rapid diagnostic testing [56], as has recently been shown during the COVID-19 pandemic.

### 4.6. Could COVID-19 Be the Silver Lining?

Over the past two years, in the midst of the COVID-19 pandemic, the general public has been flooded with information about this new virus, its management, and vaccination. People are now equipped with multitudes of information sources and the understanding that there currently is no effective treatment for this viral infection. Could this be a silver lining of these unprecedented times? Has this pandemic succeeded at increasing awareness and knowledge on the difference between viral and bacterial infections, and the understanding that antibiotics are not effective against viruses? Data from early in the pandemic show a sharp decrease in antibiotic consumption, although this is likely the result of a decline in doctor visits. As society reopens, the pandemic’s impact on health-seeking behaviour is worth investigating, particularly since visiting a physician is more likely to yield an antibiotic prescription [15]. One could also posit that awareness among patients and healthcare professionals should have improved, although it is yet to be explored in this setting.

One major drawback for RTI management in this setting was the lack of available diagnostic tools in the community. Introduction of rapid diagnostic testing in this setting has however previously been received with hesitation by GPs, primarily due to lack of human resources, infrastructure, and cost of testing [5]. Now that COVID-19 has pushed communities towards regular, and often mandated even free-of-charge testing, it will be interesting to see whether rapid diagnostic testing will become increasingly available, and at what cost, in the management of other RTIs. Concurrently, there is a need to further explore whether the public and medical professionals’ acceptance of diagnostic testing for RTIs will be sustained.

### 4.7. Strengths and Limitations

Our study was unique in that data triangulated views from three key stakeholders, giving greater in-depth understanding of GPs’, pharmacists’, and parents’ own experiences, as well as their interaction with one another. However, this study had some limitations. It was restricted to parents with tertiary-level education and so their views are limited to a particular demographic and socioeconomic group. Additionally, FGDs were held pre-pandemic and at different time points, therefore contextual changes may have impacted responses. However, as far as we are aware, between the two data collection points, there were no major policy changes that might have impacted stakeholder awareness or behaviour. Nevertheless, changes in perceptions and behaviour post-pandemic are worth exploring.

## 5. Conclusions

Our study revealed that while antibiotic use and misuse, as well as knowledge and awareness was perceived to have improved in Malta, there is still a lot of room for improvement. Fortunately, all stakeholders indicated a willingness to be involved in various initiatives to combat ABR. Addressing ABR is complex however, and will require multifaceted, targeted initiatives that address all stakeholders [59]. They must also consider contextual and cultural factors [16,60,61], and the complex interactions between healthcare professionals, patients, and the healthcare system [12].

## Figures and Tables

**Table 1 antibiotics-11-00661-t001:** Focus group discussion participation and participants’ demographic characteristics.

FGD	Stakeholders	No. of Participants	Sex		Age Range (Years)
			Male	Female	
**1**	General practitioners	6	5	1	50–70
**2**	General practitioners	2	1	1	41–59
**3**	Pharmacists	7	0	7	25–44
**4**	Pharmacists	11	3	8	25–56
**5**	Pharmacists	6	2	4	25–65
**6**	Parents	5	3	2	37–43
**7**	Parents	5	2	3	30–46
**8**	Parents	8	0	8	29–40

## Data Availability

Data that support these findings can be made available by the corresponding author upon reasonable request.

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
