# Peer review of "General Practitioners’, Pharmacists’ and Parents’ Views on Antibiotic Use and Resistance in Malta: An Exploratory Qualitative Study"

_antibiotics, 2022, doi:10.3390/antibiotics11050661_

Round 1

Reviewer 1 Report

Dear authors,

the study address an important topic- antimicrobial resistance.

Comments:

The study presents GP, pharmacists and parents’ opinion about antimicrobial use and misuse. The study describes different personal views on this important topic that should be considered when planning different strategies for improvement.

General comments: The article includes very small sample size that cannot represent population's general opinion or views. It is not clear how was sampling performed and the participants could be biased.

Three groups are very different among themselves, it is not clear how participants were invited- maybe only GP interested in AMR decided to participate or there could be other reasons for participating related to special point of view. Same is with parents - this is a special population and since it was collected with snowball sampling method the group is a representative sample of this population group. Based on this disscusions conclusions about improvment can not be draw.

Some abbreviations are not explained – FGD.

Usually, AMR for antimicrobial resistance is used as abbreviation, not ABR (antibiotic resistance), the authors use AMR in discussion only, otherwise they use ABR in the text.

In Result section:

It is not clear what were the question asked or how were the topics discussed among the GP, parents, and pharmacist?  The answers are unstructured and very common, although they address many problems that influence antibiotic prescribing culture in many countries (sick leave, lack of proper patient education etc.).

Delayed antibiotic prescribing- I am not sure if the authors understand this policy right. Delayed antibiotic prescribing means that the doctor (not patient!) delay decision about antibiotic therapy for 48 hours (such as in case of AOM). To my opinion shifting the responsibility to the patient to decide if he needs antibiotics or not is just wrong. For DAP the doctor must recheck the patient after 48 h and usually this is a problem in countries with low number of GP. Majority of infections just need the right diagnosis at the beginning (rapid lab test, rapid good microbiology tests are very helpful in decision making) and the right treatment (viral vs bacterial infection)

Over the counter antibiotic dispensing- antibiotic should not be available for selling, multi-faceted approach with interventions and policies are needed. Living on a small island is a good opportunity, not an excuse. It is important to address this issue

The GP- pharmacist relationship:

It is very good thar pharmacist are involved in AMR, they can have an important role.

Covid-19 was unfortunately a major driver of antimicrobial resistance in hospitals, whereas rapid molecular diagnostic in case of URTI was not related to lower antimicrobial prescription in one study among paediatric population. Although I do agree that rapid diagnostic is one of the important tools in antimicrobial stewardship.

Discussion:

The authors conclude that the knowledge about AMR has improved, which cannot be drawn from the presented data. Otherwise, the study addresses an important topic in populations where antibiotic stewardship should begin- GP and patients.

Author Response

Comments and Suggestions for Authors: The study address an important topic- antimicrobial resistance. The study presents GP, pharmacists and parents’ opinion about antimicrobial use and misuse. The study describes different personal views on this important topic that should be considered when planning different strategies for improvement.

General comments: The article includes very small sample size that cannot represent population's general opinion or views. It is not clear how was sampling performed and the participants could be biased. Three groups are very different among themselves, it is not clear how participants were invited- maybe only GP interested in AMR decided to participate or there could be other reasons for participating related to special point of view. Same is with parents - this is a special population and since it was collected with snowball sampling method the group is a representative sample of this population group. Based on this discussions conclusions about improvement cannot be draw.

Response: Thank you for taking the time to review our paper. We conducted an exploratory qualitative study to further understand the views of three stakeholders; the intention with this qualitative research study, was not to achieve generalizability but rather a more in-depth understanding of the challenges and behaviors in this context. We do acknowledge in our limitations the fact that parents interviewed all had tertiary education, which further limits generalizability to members of the public with a different demographic. The study nevertheless sheds light on the public’s behavior and views on certain practices.

We have gone ahead and clarified how we went about sampling the participants in the methods [lines 93-115]. As you point out, we agree that it is likely that we attracted persons who were more interested in the topic for various reasons. However, this is should not discount their views; it still helped identify gaps and areas for improvement that we believe to be important to consider for future initiatives.

Some abbreviations are not explained – FGD.

Response: Thank you for catching this. FGD (focus group discussion) written in full the first time it appears in text [lines 85-86].

Usually, AMR for antimicrobial resistance is used as abbreviation, not ABR (antibiotic resistance), the authors use AMR in discussion only, otherwise they use ABR in the text.

Response: Thank you for your comment. We have intentionally focused our paper on antibiotic resistance (ABR) and not antimicrobial resistance (AMR) because the inquiry is on antibiotic use (and thus antibiotic resistance as a consequence) and not antimicrobial use which would encompass all antimicrobial agents, e.g., antibiotics, antivirals, and antifungals. We have corrected AMR in the discussion to ABR [line 444].

Results: It is not clear what were the question asked or how were the topics discussed among the GP, parents, and pharmacist?  The answers are unstructured and very common, although they address many problems that influence antibiotic prescribing culture in many countries (sick leave, lack of proper patient education etc.).

Response: Thank you for your comment. We have now included a brief summary of the topics discussed during the focus groups [lines 89-92] and attached all the FGD guides as supplementary materials.

Delayed antibiotic prescribing- I am not sure if the authors understand this policy right. Delayed antibiotic prescribing means that the doctor (not patient!) delay decision about antibiotic therapy for 48 hours (such as in case of AOM). To my opinion shifting the responsibility to the patient to decide if he needs antibiotics or not is just wrong. For DAP the doctor must recheck the patient after 48 h and usually this is a problem in countries with low number of GP. Majority of infections just need the right diagnosis at the beginning (rapid lab test, rapid good microbiology tests are very helpful in decision making) and the right treatment (viral vs bacterial infection)

Response: We wish to point out that we do not report our views on delayed antibiotic prescribing nor our own practices in this paper but that of the participants. Nevertheless, we do wish to clarify that delayed antibiotic prescription practices do vary widely on the context and by physician, and there is an abundance of literature indicating the various methods used including: (i) providing an antibiotic prescription and instructing patients to use it after a given time frame if symptoms persist or worsen (patient-led strategy), (ii) post-dating (or forward-dating) the prescription so that patients cannot purchase it before a specified date, (iii) instructing patients to return to collect the prescription at a later date if needed, or (iv) requesting that the patient call the clinic/practitioner to issue a prescription should certain criteria be met (McDermott L, et al. (2017) Qualitative interview study of antibiotics and self-management strategies for respiratory infections in primary care).

This is also the case in Malta, where doctors delayed antibiotic prescribing practices vary widely, with some choosing to ask the patient to return if an antibiotic is not warranted at the first consultation and others giving delayed antibiotic prescriptions to the patient, and then it is ultimately up to the patient to decide whether to get the prescription dispensed once it becomes valid. Without strict guidelines and better standardization, we agree that this method can result in unnecessary antibiotic consumption and must therefore be improved, as we now recommend in the discussion: “Clearer guidance on DAP and standardization of practices is therefore highly suggested in this context.” [lines 428-429].

Shifting responsibility onto patients is definitely a controversial issue, as we found in this study. Pharmacists interviewed did not all agree with shifting decision-making onto the patient, and of particular note was the hesitancy from parents to make the decision whether to start antibiotic therapy for their children without first re-consulting a doctor [lines 211-220].

We do agree that the right diagnosis must be made from the start and that doctors should be able to differentiate between viral and bacterial infections. However, one must acknowledge the limitations and barriers faced by physicians and work towards addressing them. Rapid testing (e.g., strep tests) is currently unavailable and the country lacks the necessary infrastructure for testing in the community. Delayed antibiotic prescription can therefore be a good compromise in such a context, and a means to reduce unnecessary antibiotic consumption, as has been suggested in other studies (de la Poza Abad M, et al. (2016) Prescription strategies in acute uncomplicated respiratory infections: a randomized clinical trial; Little P, et al. (2014) Delayed antibiotic prescribing strategies for respiratory tract infections in primary care: pragmatic, factorial, randomised controlled trial.; Little P, et al. (2005) Information leaflet and antibiotic prescribing strategies for acute lower respiratory tract infection: a randomized controlled trial.; Dowell J, et al (2001). A randomised controlled trial of delayed antibiotic prescribing as a strategy for managing uncomplicated respiratory tract infection in primary care.; Little P, et al. (2001) Pragmatic randomised controlled trial of two prescribing strategies for childhood acute otitis media.; Cates C. (1999) An evidence based approach to reducing antibiotic use in children with acute otitis media: controlled before and after study.; Little P, et al. (1997) Open randomised trial of prescribing strategies in managing sore throat.)

Over the counter antibiotic dispensing- antibiotic should not be available for selling, multi-faceted approach with interventions and policies are needed. Living on a small island is a good opportunity, not an excuse. It is important to address this issue.

Response: We of course agree that antibiotics should not be dispensed without a prescription. In Malta, however, reports indicate that this practice is uncommon, although participants believed that this practice has not stopped completely. It is certainly an area that we should continue to monitor and work on in this setting. We have added a short point on this in the discussion [lines 433-434].

The GP- pharmacist relationship: It is very good thar pharmacist are involved in AMR, they can have an important role.

Response: We agree and hope that future efforts in Malta will involve pharmacists more, as we recommend in the manuscript text: “Pharmacists could also play a more active role in screening and rapid diagnostic testing, as has recently been shown during the COVID-19 pandemic.” [lines 475-476].

Discussion: The authors conclude that the knowledge about AMR has improved, which cannot be drawn from the presented data. Otherwise, the study addresses an important topic in populations where antibiotic stewardship should begin- GP and patients.

Response: Thank you for your comment. We agree that we cannot confirm that knowledge about AMR has improved and therefore adjusted our conclusion to read: “Our study revealed that while antibiotic use and misuse, as well as knowledge and awareness was perceived to have improved in Malta, there is still a lot of room for improvement.” [lines 512-514].

Reviewer 2 Report

The manuscript is clearly and well-written and interesting to read. I believe it will be of interest to other researchers working within antibiotic prescribing/resistance internationally. It reports findings from a qualitative study, which was an appropriate methodology. My comments below mostly aim to help the authors include a few more details for transparent reporting of the methods, and I encourage them also to consult an appropriate reporting checklist and add additional information as online supplementary materials (e.g. topic guides, checklist). The main limitation (although acknowledged) seems to me that the study was conducted a few years ago and a lot has likely changed in perceptions of infection management/prescribing. Hence, I suggest that the authors use a past tense when referring to past studies and findings as the current/recent situation might be different already.  

Abstract

  • Conclusion – it is somewhat inaccurate to conclude that ‘antibiotic use and misuse, and knowledge and awareness, have improved’ – first, it suggests that ‘misuse… improved’; second, the study didn’t explore change over time so it would be more accurate to state, e.g., that ‘antibiotic use, and knowledge and awareness of [ABR?] were perceived to have improved in recent time/years…
  • Last sentence – repeating ‘indicates’ and ‘indicated’

Introduction

  • Line 33 – full stop missing before ABR?
  • Line 46 –please add that this estimation related to outside of northern Europe and north America and to prescribing pre-2011 (past tense should be used instead of ‘are’).
  • I think a past tense would be better in the introduction, especially when referring to past studies. For example, reference number 7 is from 2018 – prescribing rates (e.g. line 56-7) and perceptions (line 59-60) might have changed since then.

Methods

  • Line 82 – please write FGDs in full on first use
  • Please explain how participants were recruited (e.g. how was the study advertised / how participants found out about the study), and what criteria were used for the purposive sampling?
  • Why were focus groups conducted (rather than e.g. individual interviews)?
  • Is there any reason why parents of children under 12 years were included (rather than e.g. adults with recent experience of RTIs)?
  • Were the focus groups conducted in-person/face-to-face? In what venue (e.g. university/ community etc.)?
  • It would be good to have a brief summary of the topics discussed in the focus groups, and perhaps topic guides as a supplementary document.

Results

  • ‘findings revealed’ – I’d suggest it is more accurate to say that ‘categories were identified’ or ‘findings were organised into, and are presented under, the three categories’, thus acknowledging an active role of researchers who analyse and interpret the data.
  • Quote in lines 160-2 – I’m not sure it illustrates well the point made in the sentence immediately before. It seems to suggest that older doctors display more habitual prescribing, whereas the sentence states that GPs were seen as overprescribing irrespective of age. Also the sentence is about parents’ views, whereas the quote show pharmacist’s view. I’d suggest replacing the quote or amending the sentence.

Discussion

  • Line 354 - ‘is’ missing and ‘with’ unnecessary at the end of the line?
  • Line 359 – perhaps more accurate to say ‘perceived improvement’ than ‘observed’.
  • Line 467 – why were parents only with tertiary-level education included? This should be explained in the methods, together with approaches to recruitment and sampling.
  • Line 475-6 – the same comment as in the abstract conclusion about improvement in antibiotic use and misuse.

Overall, I would also recommend checking the reporting with a relevant reporting checklist for qualitative studies, such as COREQ or SRQR.

Author Response

Thank you for taking the time to thoroughly critique our manuscript. Kindly find our point-by-point responses to all comments below.

The manuscript is clearly and well-written and interesting to read. I believe it will be of interest to other researchers working within antibiotic prescribing/resistance internationally. It reports findings from a qualitative study, which was an appropriate methodology. My comments below mostly aim to help the authors include a few more details for transparent reporting of the methods, and I encourage them also to consult an appropriate reporting checklist and add additional information as online supplementary materials (e.g., topic guides, checklist). The main limitation (although acknowledged) seems to me that the study was conducted a few years ago and a lot has likely changed in perceptions of infection management/prescribing. Hence, I suggest that the authors use a past tense when referring to past studies and findings as the current/recent situation might be different already.

Response: Thank you for taking the time to thoroughly review our manuscript. We have now included more details in the methods for more transparent reporting and added all our FGD guides as supplementary materials. We have also consulted COREQ as recommended. With regards to using the past tense, we have changed the tenses where deemed appropriate.

Abstract

  • Conclusion – it is somewhat inaccurate to conclude that ‘antibiotic use and misuse, and knowledge and awareness, have improved’ – first, it suggests that ‘misuse… improved’; second, the study didn’t explore change over time so it would be more accurate to state, e.g., that ‘antibiotic use, and knowledge and awareness of [ABR?] were perceived to have improved in recent time/years…
    • Response: Thank you for pointing this out to us. We have now changed our conclusion in both the abstract and the main text to read: “While antibiotic use and misuse, and knowledge and awareness, were perceived to have improved in Malta, our study suggests that while stakeholders indicated willingness to drive change, there is still much room for improvement.”
  • Last sentence – repeating ‘indicates’ and ‘indicated’
    • Response: Thank you for drawing our attention to this. We have now changed indicates to suggests instead [line 25].

Introduction

  • Line 33 – full stop missing before ABR?
    • Response: Thank you for noting this error on our part. Full stop added [line 33].
  • Line 46 – please add that this estimation related to outside of northern Europe and north America and to prescribing pre-2011 (past tense should be used instead of ‘are’).
    • Response: We have added this to the introduction and used the past tense in this sentence [lines 46-48].
  • I think a past tense would be better in the introduction, especially when referring to past studies. For example, reference number 7 is from 2018 – prescribing rates (e.g. line 56-7) and perceptions (line 59-60) might have changed since then.
    • Response: We have changed the tenses to the past tense where deemed appropriate.

Methods

  • Line 82 – please write FGDs in full on first use
    • Response: Thank you for catching this. FGD (focus group discussion) written in full the first time it appears in text [lines 85-86].
  • Please explain how participants were recruited (e.g., how was the study advertised / how participants found out about the study), and what criteria were used for the purposive sampling?
    • Response: Thank you for your comment. We have now added more sampling and recruitment details in the methods [lines 91-115].
  • Why were focus groups conducted (rather than e.g., individual interviews)?
    • Response: Thank you for your query. We primarily sought to carry out focus group discussions and not individual interviews because we did not have the time or resources at the time to conduct individual interviews with so many participants. For this reason, we opted for FGDs. We also believed that it might be of value to bring healthcare professionals in particular, to discuss the topic; that it could possibly add more nuance to our data by having them discuss as a group. We recognize however that it is possible that some participants may have been less comfortable to voice their opinion in a group, however the moderator did their utmost to ensure that everyone’s views got heard and shared with the group. We also did not mix the groups, i.e., we did not hold FGDs with a mixture of the different stakeholders but rather kept them separate in order not to create any power imbalances.
  • Is there any reason why parents of children under 12 years were included (rather than e.g., adults with recent experience of RTIs)?
    • Response: From a sampling perspective, it was more convenient at the time to include parents in our study since we had already established connections with schools and were able to identify parents through those schools. We also hypothesized that children are highly likely to receive antibiotics unnecessarily and wished to capture the views of their guardians, in this case their parents.
  • Were the focus groups conducted in-person/face-to-face? In what venue (e.g., university/ community etc.)?
    • Response: They were held in-person at a hotel (GPs) and the local governmental hospital (pharmacists and parents) as has now been clarified in the methods. [lines 100-102 and 113-115].
  • It would be good to have a brief summary of the topics discussed in the focus groups, and perhaps topic guides as a supplementary document.
    • Response: Thank you for this recommendation. We have gone ahead and added a brief summary of what was discussed in the focus groups [lines 89-92] and added all the guides as supplementary materials as suggested.

Results

  • ‘findings revealed’ – I’d suggest it is more accurate to say that ‘categories were identified’ or ‘findings were organised into, and are presented under, the three categories’, thus acknowledging an active role of researchers who analyse and interpret the data.
    • Response: Thank you for your suggestion. We have tweaked the sentence to read: “Our findings are organized and presented across three overarching categories: (1) antibiotic use and ABR in Malta from the perspective of three stakeholder groups, (2) influence of interpersonal relationships among patients, GPs, and pharmacists on antibiotic use, and (3) solutions for action – tackling ABR in Malta.” [lines 135-139].
  • Quote in lines 160-2 – I’m not sure it illustrates well the point made in the sentence immediately before. It seems to suggest that older doctors display more habitual prescribing, whereas the sentence states that GPs were seen as overprescribing irrespective of age. Also the sentence is about parents’ views, whereas the quote show pharmacist’s view. I’d suggest replacing the quote or amending the sentence.
    • Response: Thank you for noting this. We have changed the order of that paragraph so that the quote now appears before the sentence on parents’ views [lines 186-188].

Discussion

  • Line 354 - ‘is’ missing and ‘with’ unnecessary at the end of the line?
    • Response: Thanks for catching this error! We have made the change [line 382].
  • Line 359 – perhaps more accurate to say ‘perceived improvement’ than ‘observed’.
    • Response: Thank you for your suggestion. We have changed observed to perceived [line 392].
  • Line 467 – why were parents only with tertiary-level education included? This should be explained in the methods, together with approaches to recruitment and sampling.
    • Response: We have further explained our snowball sampling in the methods [lines 91-115]. Regarding why parents with tertiary-level education only were included, this was not intentional but we believe the result of our sampling method attracting people within the same social circles and with similar backgrounds.
  • Line 475-6 – the same comment as in the abstract conclusion about improvement in antibiotic use and misuse.
    • Response: We have made changes to the conclusion as previously suggested.